# Pharmacokinetic-Pharmacodynamic Modelling of Systemic IL13 Blockade by Monoclonal Antibody Therapy: A Free Assay Disguised as Total

**DOI:** 10.3390/pharmaceutics13040519

**Published:** 2021-04-09

**Authors:** John Hood, Ignacio González-García, Nicholas White, Leeron Marshall, Vincent F. S. Dubois, Paolo Vicini, Paul G. Baverel

**Affiliations:** 1Clinical Pharmacology and Quantitative Pharmacology, AstraZeneca, Cambridge CB21 6GH, UK; ignacio.gonzalez@astrazeneca.com (I.G.-G.); nicholas.white@astrazeneca.com (N.W.); leeronmarshall@hotmail.com (L.M.); vincent.dubois@astrazeneca.com (V.F.S.D.); pvicini@live.com (P.V.); paul.baverel@roche.com (P.G.B.); 2Salford Royal Foundation Trust, Salford M6 8HD, UK; 3Confo Therapeutics, 9052 Ghent, Zwijnaarde, Belgium; 4Roche Pharma Research and Early Development, Clinical Pharmacology, Pharmaceutical Sciences, Roche Innovation Center Basel F. Hoffmann-La Roche Ltd., CH-4070 Basel, Switzerland

**Keywords:** pharmacokinetic-pharmacodynamic modelling, IL13, assay, monoclonal antibodies, population analysis, dose-exposure-response, NONMEM

## Abstract

A sequential pharmacokinetic (PK) and pharmacodynamic (PD) model was built with Nonlinear Mixed Effects Modelling based on data from a first-in-human trial of a novel biologic, MEDI7836. MEDI7836 is a human immunoglobulin G1 lambda (IgG1λ-YTE) monoclonal antibody, with an Fc modification to reduce metabolic clearance. MEDI7836 specifically binds to, and functionally neutralizes interleukin-13. Thirty-two healthy male adults were enrolled into a dose-escalation clinical trial. Four active doses were tested (30, 105, 300, and 600 mg) with 6 volunteers enrolled per cohort. Eight volunteers received placebo as control. Following single subcutaneous administration (SC), individual time courses of serum MEDI7836 concentrations, and the resulting serum IL13 modulation in vivo, were quantified. A binding pharmacokinetic-pharmacodynamic (PK-PD) indirect response model was built to characterize the exposure-driven modulation of the target over time by MEDI7836. While the validated bioanalytical assay specification quantified the level of free target (i.e., a free IL13 assay), emerging clinical data suggested dose-dependent increase in systemic IL13 concentration over time, indicative of a total IL13 assay. The target time course was modelled as a linear combination of free target and a percentage of the drug-target complex to fit the clinical data. This novel PK-PD modelling approach integrates independent knowledge about the assay characteristics to successfully elucidate apparently complex observations.

## 1. Introduction

Asthma is one of the major noncommunicable diseases, and was estimated to have affected more than 300 million people globally in 2016 [1]. Asthma is characterized by hyperresponsiveness to specific and non-specific stimuli, chronic pulmonary eosinophilia, upregulated serum IgE, and excessive airway mucus production. Asthma is believed to be mediated by CD4+ T-lymphocytes, which produce type 2 cytokines including IL4 and IL5. IL13 has been implicated in development and expression of airway hyperresponsiveness, with IL13 alone shown to be sufficient to induce responses in murine models [2]. As IL13 is a validated target in driving the allergic asthma phenotype, IL13 neutralization had attractive therapeutic potential [3].

The standard of care for allergic asthma consists of inhaled corticosteroids and short- and long-acting bronchodilators, which control asthma symptoms in the majority of patients, complemented by anti-IL4 and anti-IL5 pathway therapies [4]. Over forty percent of asthma patients still have trouble controlling symptoms, with a greater risk of exacerbation and increased mortality [4]. Novel, long-term, preventative treatments for asthma are a necessity. A range of monoclonal antibody (mAb) therapies targeting components of the inflammatory pathway have been developed and are available to patients. Approved therapies targeting the Th2 pathway include: benralizumab targets the IL5 receptor blockade while mepolizumab targets IL5, omalizumab targets IgE while dupilumab targets IL4Rα, Anti-IL13 therapies (lebrikizumab and tralokinumab) have been shown to reduce the risk of exacerbations, particularly in patients with high serum periostin or high fraction of exhaled nitrous oxide (FeNO), and have been shown to increase forced expiratory volume (FEV1) in patients compared to the placebo [4,5,6]. Anti-IL13 treatment has been associated with a greater improvement in Asthma Quality of Life Questionnaire (AQLQ) scores, with a decrease in the use of rescue medication, but has not consistently confirmed an effect on asthma exacerbations [6,7].

MEDI7836 was developed as a best-in-class back-up program of tralokinumab when the latter and lebrikizumab entered in Phase 3. Reducing dosing frequency of tralokinumab and lebrikizumab, given bi-weekly and monthly respectively, while maintaining asthma control in patients, presented practical and societal incentives that motivated the development of MEDI7836. Two differentiation attributes were engineered: increased affinity for IL13 (from 58 pM for tralokinumab to 5 pM for MEDI7836, measured by Biacore T100 [8]) and a mutation in the Fc fragment of MEDI7836 to optimize its interaction with neonatal Fc receptor (FcRn) and consequently extend its serum half-life. The M252Y/S254T/T256E AA substitution (YTE) was shown to improve FcRn binding by 10-fold, leading to a reduction in MEDI7836 clearance compared to a standard IgG [9]. MEDI7836 low clearance combined with the increased affinity for IL13 compared to tralokinumab aimed at achieving a more prolonged target suppression for a similar binding molar equivalent of drug at the site of action. Consequently, MEDI7836 was designed to have a greater potency and a longer dosing interval than tralokinumab, with a projected therapeutic efficacious dose range compatible with a ≥6 weeks interval dose regimen based on a theoretical IL13 target suppression pharmacokinetic and pharmacodynamic (PK-PD) model.

A limitation of antibody therapies is that proteins are inherently immunogenic [10]. Following the initial dose of an antibody therapy, there is the potential for patients to develop anti-drug antibodies (ADAs), which can have a number of consequences. ADAs can enhance clearance and cause serum drug levels to drop to sub-therapeutic levels, or neutralize the activity of the drug and cause a loss of clinical efficacy [10]. ADAs can also contribute to injection-site reactions, thromboembolic events, and serum sickness. Quantification of ADAs is an essential step in clinical trials of biologics. Both tralokinumab and lebrikizumab demonstrated low ADA incidence rate and ADA were transient in nature [6,7].

The capability to reliably and accurately quantify drug and target concentrations from biological samples, including blood, urine, and tissues, is vital to determine the properties of candidate drugs with a targeted mechanism [11]. When developing bioanalytical assays, it is important to determine what the assay measures; multiple forms of drug, target, and drug-target complex are likely to be present in samples [11]. Depending on whether the assay detects only unbound target, unbound drug, or detects drug-target complex, the assay is described as ‘free’ (if it only detects unbound analyte), or ‘total’ (if it detects bound and unbound analyte). Generally, ‘free’ assays are more valuable for bioanalysis of drug targets, as they demonstrate the extent of suppression of target by the drug. Bioanalytical assays are then characterized as ‘free’ or ‘total’ during their development and validation. Humanized antibody therapies require specific anti-idiotype antibodies to be reliably quantified in human samples; general anti-human IgG antibodies will detect endogenous and exogenous antibodies [11].

The work presented here involved analysis of a sequential PK-PD data set from a first in human (FIH) trial of MEDI7836, a novel anti-IL13 mAb with a YTE mutated Fc fragment for half-life extension [9,12]. The data consisted of individual time courses of MEDI7836 following SC doses, together with the time course of the resulting IL13 modulation in vivo. The intent of the analysis was to establish a PK-PD framework able to integrate known elements of the target biology to explain the observed data and to increase our mechanistic understanding of the drug exposure–target modulation relationship in vivo. We postulate that the PK-PD modelling approach can provide a mechanistic hypothesis framework to rationalize the counterintuitive clinical data generated by a free target engagement assay following MEDI7836 administration in healthy volunteers, and demonstrate its value to disentangle the contribution of MEDI7836 serum concentrations and ADA in the target modulation responses observed in vivo over time.

## 2. Materials and Methods

### 2.1. Clinical Trial Design

Study NCT02388347 was a randomized, blinded, placebo-controlled study to evaluate the safety and tolerability of single-ascending SC doses of MEDI7836 in healthy adult male subjects. The study was conducted at a single site in the UK. The secondary objective of the study was to investigate the PK and immunogenicity of MEDI7836, and the effect of MEDI7836 on serum IL13 was an exploratory objective.

Four dosing cohorts of MEDI7836 at doses of 30, 105, 300, and 600 mg, or placebo included a total of 32 subjects (24 subjects received MEDI7836, 8 subjects received placebo). Individual subjects were randomized at each dose level in a 6:2 ratio to receive MEDI7836 or placebo.

Subjects were in the study for up to Day 281, including a screening period of up to 28 days, a single-dose treatment day (during a 4-day inpatient period), and a 280 day post-treatment follow-up period due to the anticipated extended half-life of MEDI7836. Subjects returned to the study site on Days 6, 8, 10, 15, 29, 43, 57, 85, 113, 169, 225, and lastly 281 for a safety follow-up visit.

A summary of the trial design and sampling times is shown in Figure 1.

### 2.2. Bioanalysis

Quantification of free MEDI7836 human serum was done using a Gyrolab-based sequential flow-through immunoassay method (Gyros Protein Technologies, Uppsala, Sweden). The MEDI7836 quantification assay utilized reagents generated by MedImmune Ltd. via hybridoma; a biotinylated anti-MEDI7836 mAb capture reagent, and an anti-YTE mAb detection reagent conjugated to Alexa 647 were used.

Anti-drug antibodies (ADA) to MEDI7836 were detected using an electrochemiluminescence immunoassay method using a Meso Scale Discovery (MSD) Sector Imager 6000 (SI6000) (Meso Scale Diagnostics, Rockville, MD, USA). The ADA screening assay was based on a bridging immunoassay format; positive control samples, screening samples, and negative control samples were incubated in the presence of biotinylated MEDI7836 and a ruthenylated MEDI7836 master mix solution. The complex was then added to a pre-blocked, streptavidin coated MSD plate and further incubated. Following a wash step, the MSD read buffer was added and the plate loaded into the MSD SI6000 for analysis.

Quantification of IL13 was done using a proprietary Singulex human IL13 immunoassay kit (Singulex/Merck Millipore, Watford, Hertfordshire, UK). Concentrations of IL13 in serum were measured using a qualified bioanalytical Singulex Erenna method.

### 2.3. Data Description

The integrated model was built on MEDI7836 (PK) and IL13 (PD) serum concentration data taken from the Phase 1 clinical trial NCT02388347. All individual measures of exposures were pooled along with MEDI7836 dosing information and covariates, including baseline demographics, immunogenicity status, and routine safety lab parameters. Individual time-course representation of the PK and PD measurements for each cohort of subjects receiving SC MEDI7836 doses are included in Figure 2 and Figure 3, respectively. Patient demographic information is included in Table 1. In total, an average of 13.9 PK samples per individual out of 24 individuals, and an average of 14.9 PD samples per individual out of 32 individuals compose the analysis dataset. Based on Figure 3, observed IL13 increased following administration of MEDI7836. This trend in the data was inconsistent with predicted free IL13, which should have decreased following administration of MEDI7836 due to its neutralizing activity and high affinity for IL13 and the low expression levels of the target expected in healthy volunteers. During initial graphical exploration, it was noted that one patient in the control arm had a PD profile suggestive of dosing with MEDI7836 and a patient in the 600 mg arm had PD data suggestive of belonging in the control arm. Since these time courses were not consistent with the time course response of subjects in the same cohort, and the possibility they could have been swapped in error could not be excluded, these data were excluded from the analysis.

### 2.4. Modelling Methodology

The clinical dataset comprising of individual PK and PD time profiles and their respective covariate information at baseline was created to perform a population PK-PD modelling analysis following the basic concepts of non-linear mixed-effect model building [13]. Dependent variables were converted from µg/mL to nM for PK (Figure 2), and from pg/mL to nM for PD (Figure 3) to ensure binding of MEDI7836 with its target IL13 was done at the molar scale given the difference in molecular mass (150 KDa for MEDI7836 vs. *circa* 10 KDa for IL13). Concentrations for PD were log-transformed to ease the modelling, due to the wide dynamic range of IL13 data. All modelling was performed using a non-linear mixed-effect modelling methodology as implemented in the software NONMEM [13,14,15]. Stochastic approximation of expectation maximization (SAEM) and Monte Carlo importance sampling (IMP) were employed to maximize the likelihood of model parameters with respect to the observed data. Using this approach, inter-individual variability of PK and PD parameters were estimated from the data. The model was built in a stepwise manner. First, the PK data were used to develop the structural PK model, investigating a range of inter-individual and residual variability models (IIV and RSV, respectively). Model selection was based on minimizing objective function value (OFV) [13,16,17]. The PK model predictive adequacy was then validated using standard visual predictive check with 500 simulations while uncertainty of PK parameter estimates was obtained by bootstrapping the original dataset with 500 replicates. The PK parameter estimates were then fixed to permit PD model development based on OFV minimization for nested models [17]. Simultaneous estimation of PK and PD parameters was not attempted. The statistical level for inclusion of model parameters was set to *p* < 0.05 for structural parameters and stochastic model components. For covariate analysis, the same statistical level was used.

### 2.5. Structural Model

Several structural models were evaluated to fit the PK data. Bioavailability parameter F was not estimated since no intravenous data of MEDI7836 were available so PK parameters remained apparent (e.g., CL/F for systemic clearance of MEDI7836). The PK-PD model structure consisted of a multiple-compartment PK-PD binding model relating the concentration of MEDI7836 in serum with target suppression in the systemic circulation and following a turnover indirect response (see Figure 4). An earlier version of the model was briefly described in [18].

The binding (K_on_) and dissociation (K_off_) rates of MEDI7836:IL13 complex formation were fixed based on affinity estimates generated during molecule characterization (K_on_; 138.24 nM^−1^day^−1^, K_off_; 0.69 day^−1^). IL13 expression at baseline and turnover rate were estimated according to an IL13 constant turnover model with a single compartment, a production rate (K_in_), and an elimination rate (K_out_) assumed similar to the unbound mAb MEDI7836. This assumption holds due to the soluble nature of IL13. The IL13:MEDI7836 complex is produced in the central compartment. IL13 estimates of RSV were expressed as coefficient of variation (%CV) for proportional error model components [19].

### 2.6. Stochastic Model

MEDI7836 inter-individual variability (IIV) was described through variability in selected PK and PD parameters, as supported by data. The stochastic components of the PK-PD model were assumed to follow a log-normal profile and were described by Equation (1):(1)$$\theta_{i}=TV\theta \cdot e^{\eta_{i}},$$
where *TVθ* represents the mean estimate (or typical value) of PK or PD parameter θ in the population (e.g., CL/F or K_out_), and *η_i_* is the subject-specific random effect for the i-th individual, assumed to have a normal distribution with a mean of zero and a variance of ω^2^. The correlations between the variability estimate ω^2^ of each PK parameter were estimated, if supported by the data.

Different error structures of the PK-PD model (residual error defined as the difference between prediction and observation within an individual due to random variability, including assay variability or error in measurements) were tested (Equation (2)), where *yij* is the jth observation in the ith individual, *ŷij* is the corresponding model prediction, and ε_1_ and ε_2_ are parameters for combined residual error (corresponding to the proportional and additive components). Proportional residual error was used for PD (Equation (3)), where *y_ij_* is the jth observation in the ith individual, IL13*_ij_*, FR*_ij_*, and *C_xij_* are the model predictions for the jth observation of free IL13, complex fraction, and concentration of MEDI7836:IL13 complex for the ith individual, respectively, and ε_3_ is the proportional residual error. Coefficient of variation for lognormal data was estimated using the method from Elassaiss–Shaap and Heisterkamp (2013) (Equation (4) [19]):(2)$$y_{ij}={ŷ}_{ij}\cdot \left(1+\varepsilon_{1}\right)+\varepsilon_{2},$$
(3)$$y_{ij}=\log\left(\mathrm{IL}13_{ij}+{\mathrm{FR}}_{ij}\cdot C_{xij}\right)\cdot \left(1+\varepsilon_{3}\right),$$
(4)$$CV\%=\sqrt{\left(e^{\omega^{2}}-1\right)}.$$

### 2.7. Investigated Covariate Relationships

Multiple covariates consisting of subjects’ characteristics at baseline were tested for their ability to explain IIV in PK parameters. Body weight, age, race, and immunogenicity status were evaluated based on scientific interest and mechanistic plausibility. The relationship between parameters and covariates tested is shown in Equations (5) and (6), for discrete and continuous variables, respectively:(5)$$P=\theta_{tv}\cdot \left(1+\theta_{cov}\right),$$
(6)$$P=\theta_{tv}\cdot \left(1+\theta_{cov}\cdot \left(cov-cov_{median}\right)\right),$$
where *P* refers to a model parameter with a covariate inclusion supported by the data, $\theta_{tv}$ the typical value for the most represented category of a discrete covariate or at the median for a continuous covariate (cov = $cov_{median}$), and $\theta_{cov}$ is an estimate of the covariate-model parameter relationship magnitude.

### 2.8. Software

All modelling was done in NONMEM (ICON Development Solutions, Ellicott City, MD, USA; 2006) using the ADVAN13 subroutine in NONMEM, version 7.3 [14]. Perl-speaks-NONMEM (PsN [15,20]) was used for bootstrapping. Visual predictive checks and plotting were done using the dplyr (version 1.0.2) [21] and ggplot2 (version 3.3.2) [22] packages in R Studio (version 1.1.463, R Core Team; 2020, R version 4.0.0). The simulation in Figure 7 was done in SAAMII v.2.3 (The Epsilon Group, Charlottesville, VA, USA) and plotted using ggplot2 in R Studio [22].

## 3. Results

### 3.1. PK Model

The two-compartment model with linear elimination and an absorption compartment best described the concentration-time data for MEDI7836. SC administration enters via a depot compartment before diffusing to the central (blood) compartment and being distributed between the central and peripheral (tissue) compartment. The PK model was parameterized in terms of the following parameters; first-order absorption constant rate (K_a_, 0.156 1/day, 12.3% relative standard error (RSE)), apparent systemic clearance (CL/F, 0.441 L/day, 12.4% RSE), apparent volume of distribution in the observed central compartment (V_2_/F, 2.83 L, 20.9% RSE), apparent volume of distribution in the peripheral compartment (V_3_/F, 8.03 L, 11.1% RSE), intercompartmental clearance (Q/F, 0.825 L/day, 17.3% RSE). The covariate analysis revealed that ADA positive post-baseline status was statistically associated with faster MEDI7836 mAb clearance (ADA CL increase, 71.7%, 49.4% RSE) (Table 2 and Figure 4). No other covariate was found to be associated with the PK properties of MEDI7836. The impact of body weight on volume and clearance parameters was tested as a covariate but the association was not significant (*p* < 0.05).

IIV was included for CL/F, V_2_/F, V_3_/F, and Q/F (53.3% (8.88% RSE,), 72.7% (14.7% RSE), 46.2% (6.63% RSE), and 69.6% (6.95% RSE), respectively. A combined error model was used to account for the PK assay residual variability (additive RSV was 0.0546 µg/L (32.2% RSE) and the proportional RSV was 13.0% (9.43% RSE). Overall, visual predictive check (VPC) shows a good agreement between predictions and observed percentiles (Figure 5).

### 3.2. PD Model

A baseline PD model describing IL13 turnover was parameterized as K_in_ (0.0173 nmol/day, 26.7% RSE) and K_out_ (180.0 1/day, 20.6% RSE) with PK-PD complex formation with MEDI7836 based on assumed K_on_ (138.0 1/day*nM) and K_off_ (0.690 1/day) parameter set to in vitro binding estimates obtained from Biacore. Distribution and elimination of the IL13:MEDI7836 complex was best described using the same clearance as unbound MEDI7836, with a different apparent central volume of distribution (V_cx_/F, 13.6 L, 16.4% RSE). Adding a peripheral compartment for the IL13:MEDI7836 complex with the same apparent intercompartmental clearance (Q/F) and apparent volume (V_3_/F) as unbound MEDI7836 showed a statistically better fit to the data (*p* < 0.05). The schematic of the final model is shown in Figure 6.

The increase in observed IL13 suggested that the bioanalytical assay was measuring a fraction of the IL13:MEDI7836 complex, as the observed magnitude of increase in complex formation in the data was not sufficient to corroborate a full total IL13:MEDI7836 assay. Using the 105 mg dose, a simulation from the final PK-PD model was set to illustrate that a free IL13 assay would in theory show a decrease in observed IL13 concentration to less than 0.01 pM (Figure 7); however, an increase in IL13 concentration to ~5 pM was observed, which was not enough to imply a total IL13 assay that would have resulted in *circa* 100 pM. The fraction of the complex (C_x_) assumed captured by the bioanalytical PD assay was estimated to be small at 4.29% (25.3% RSE) of the total MEDI7836:IL13 predicted by the PK-PD model based on mass transfer binding. IIV was implemented for K_out_ (57.1%, 33.2% RSE), and C_x_ Fraction (154.0%, 38.7% RSE).

Figure 5 shows the VPC of the placebo and active PD time-course of MEDI7836. The VPC confirms good accuracy between predictions and observed percentiles, suggesting that the model is adequate. Bootstrapped parameter estimates showed a narrow confidence interval. All model parameters were biologically plausible.

Additional diagnostic plots providing more detail about model performance and goodness of fit are available in the Appendix A, as well as the final PK-PD model NONMEM control stream.

## 4. Discussion

A PK-PD modelling framework encapsulating the exposure-response observed following administration of MEDI7836 in healthy patients single-ascending dose trial is presented to rationalize the unexpected behaviour of a PD target engagement bioanalytical assay. Overall, PK-PD parameter estimates were all biologically plausible for a mAb directed against a soluble target. The PK of MEDI7836 did not exhibit a supra- or sub-proportional trend with dose, suggesting linear PK properties of MEDI7836 in the range of doses evaluated in this trial. A similar structural population PK model was assumed for the parent molecule tralokinumab [23] and lebrikizumab [24], suggesting broadly equivalent PK characteristics in the class of anti-IL13 mAbs. However, some discrepancies are notable, since MEDI7836 was expected to be a biosuperior with favourable PK properties compared to the first generation of anti-IL13 mAbs. The apparent central compartment volume (V_2_/F) of MEDI7836 was of similar order of magnitude to tralokinumab (2.8 compared to 2.3 L, respectively) [23]. The apparent peripheral volume (V_3_/F) is two-fold greater for MEDI7836 than tralokinumab (8.0 compared to 4.2 L, respectively [23]), suggesting a potential off-target binding or a low bioavailability after SC administration since V_ss_/F greatly exceeds the expected plasma volume in healthy volunteers (*circa* 5 L). The PK of MEDI7836 was also less favourable than lebrikizumab, with a faster CL/F (0.441 compared to 0.182 L/day, respectively), smaller V_2_/F (2.8 compared to 4.8 L, respectively), and greater V_3_/F (8.0 compared to 1.7 L, respectively) [24]. Overall, the PK of MEDI7836 was linear, but showed high level of inter-individual variability (range 40–65% for all apparent disposition parameters). The apparent clearance of MEDI7836 was greater than tralokinumab (0.441 compared to 0.255 L/day, respectively), which was unexpected for a YTE molecule [23] and could not be explained solely by a posited low bioavailability. Post-baseline ADA positive status was found to increase elimination of MEDI7836 by 71.7%. This result corroborated the observations that subjects experiencing ADA response to MEDI7836 treatment had reduced exposure levels compared to ADA negative subject at the same dose, and further substantiated the hypotheses of nonspecific binding and immune-related clearing mechanisms of MEDI7836. The absorption rate K_a_ was of similar order of magnitude as tralokinumab and lebrikizumab, suggesting no major difference the absorption profile between the 3 molecules beside the bioavailability that was not estimated for MEDI7836 due to lack of intravenous data. The PD parameter estimates were biologically plausible with a turnover half-life of IL13 in serum of 8 min. Overall, these data allowed to confirm target engagement in the systemic circulation. Free serum IL13 at baseline were estimated as K_in_/K_out_ which equates on average at 0.096 pM in healthy volunteers. These levels are aligned with a low inflammation status and demonstrate the high sensitivity of the PD assay. Model parameters were precisely estimated irrespective of PK or PD data with RSE not exceeding 40%, which is consistent with the rich sampling scheme typical of an FIH trial. The FIH design had a long follow-up period (281 days after administration of MEDI7836 single dose) and thus allowed a precise estimation of all disposition parameters, including the apparent systemic CL while PD parameters were all identifiable and well estimated since a return to baseline levels of IL13 was observed.

The main finding of this work was that the assumption that the assay measured a mixture of free IL13 and a portion of the complex was supported by the totality of the clinical data. The complex fraction parameter adequately described the observed PD data and estimated that, on average, only 5% of the complex was captured by the detection reagents of the PD assay. Based on our modelling analysis, the IL13 assay is neither free nor fully total, but a combination of the two, suggesting some interference in the capture or detection reagents of the PD assay when dealing with the clinical samples of the trial. This suggests that the samples used to develop and validate the free assays are different from the samples obtained from the individuals in the trial. 

The model was also used to confirm whether a dosing frequency of MEDI7836 given every 6 weeks would have resulted in sufficient target suppression coverage (Figure 8). The model simulations were instrumental for decision making, since the projected efficacious dose range did not match the target product profile for best-in-class potential pursued for MEDI7836, which consequently halted its clinical development (Figure 8).

MEDI7836 had an increased rate of clearance relative to that expected from an IgG with a YTE mutation [12]. Antibody clearance is driven by the interplay of FcRn interactions, target mediated drug disposition (TMDD), and by nonspecific binding [25]. The characteristics of IL13 make TMDD unlikely. Since the target IL13 is a circulating cytokine, MEDI7836:IL13 complex formation would occur in the central or peripheral (tissue) compartment with no internalization mechanisms such as for mAb targeting membrane-bound receptors. Moreover, the IL13 turnover is not sufficiently rapid and the IL13 expression level at baseline not sufficiently abundant relative to the elimination half-life and molar ratio of MEDI7836 following doses tested in this trial to justify a target-mediated disposition of free mAb. Indeed, the PK profile of MEDI7836 was typical of an IgG and not consistent with TMDD nor consistent with YTE half-life extension. As the YTE mutation was developed to optimize FcRn binding, the most likely explanation for the greater than expected clearance is non-specific binding that may have resulted from the low pM-range affinity optimization of the lead clone. Nonspecific binding of antibodies is related to their physical and chemical properties, including isoelectric point, net charge, stability and aggregation potential, post-translational modification, and the characteristics of the variable region [25]. The impact of these properties on nonspecific binding can mask or limit Fc modifications and antibody affinity, and understanding these effects can be critical to effective antibody engineering. It is likely that the comparatively large peripheral volume of MEDI7836 (8.0 L) is attributable to nonspecific binding.

Even if constant regions of an antibody are identical, differences in the variable region have been shown to have a significant effect on clearance [26]. Certain properties of variable regions can promote promiscuous off-target binding to tissues, which can accelerate the systemic clearance of antibodies [26]. Unfortunately, it is difficult to predict these off-target binding effects from sequence data, in vitro assays, or in silico modelling; hydrophobicity and average charge are not always greater in molecules showing unfavourable PK properties [26]. A key learning from this clinical trial was that more tools and experiments would have been needed to assess nonspecific binding liabilities in candidate molecules to fully interpret FIH clinical data.

There is a high probability that the fraction of total IL13 detected was linked to the bioanalytical method utilized. The IL13 detection was done using single molecule counting (SMC) on the Erenna platform from Singulex [27]. There are two potential causes for the observed IL13 measurements; firstly, a component of the anti-IL13 detection antibody could be binding to an epitope exposed on the MEDI7836:IL13 complex [28]. Secondly, nonspecific binding could be responsible for the detected fraction of MEDI7836:IL13 complex, driven by the same mechanisms affecting clearance [26]. Given the confidence intervals on the estimate of C_x_ fraction (0.0298–0.0596), the first hypothesis appears more likely. Future validations of bioanalytical assays would benefit from quantifying drug targets in the presence of drug to rule out mixed ‘free’/‘total’ assays in future.

This analysis has a number of limitations. Firstly, there was no free or total IL13 assay developed *a posteriori* to confirm or refute the hypothesis of this work. Secondly, ADA data was sparse and did not allow in depth analysis of the transient and persistent nature of ADA against MEDI7836. Moreover, the association of ADA status with PK was limited as ADA status was not entered in the model as a time-varying covariate and may have confounded or biased some of the model outcomes. Finally, the small number of patients (*n* = 32) and single SC dose did not allow us to estimate bioavailability and limited the extrapolation properties of the model for multiple doses, especially given the high immunogenic potential of MEDI7836 since ADA would not typically be expected after a single dose.

## 5. Conclusions

We described here a theoretical PK-PD model of a biosuperior mAb and its interactions with its target IL13 with a standard binding kinetic modelling and statistical framework to rationalize unexpected clinical PD, ADA, and PK data from a FIH trial. Individual time courses of MEDI7836 concentrations and resulting target suppression following SC administration IL13 of MEDI7836 were described by an indirect response model in a population approach, which described the target time course as a linear combination of free target and a percentage of the drug-target complex. While validation data indicated the bioanalytical assay quantified free IL13, the proposed model better described the observations by accounting for a relatively small percentage of MEDI7836:IL13 complex being captured by the assay. This novel PK-PD modelling approach integrates independent knowledge about the assay design and explains the observations well and can provide insight on the antibody–target interactions even when target modulation data are not immediately interpretable.

## Figures and Tables

**Figure 1 pharmaceutics-13-00519-f001:**
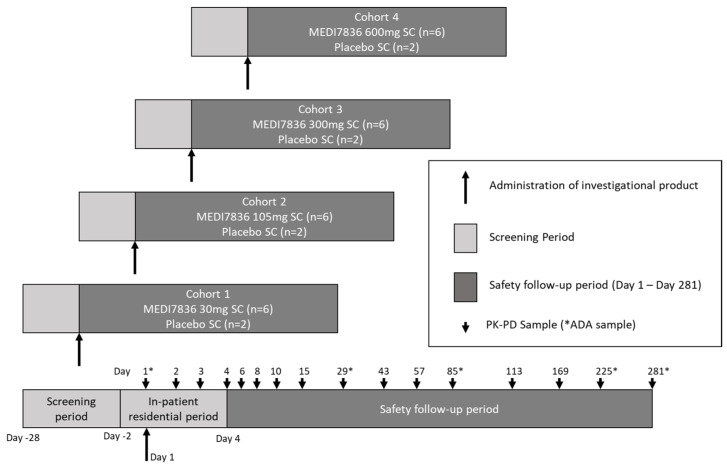
Trial design and sampling scheme for the MEDI7836 Phase 1 clinical trial.

**Figure 2 pharmaceutics-13-00519-f002:**
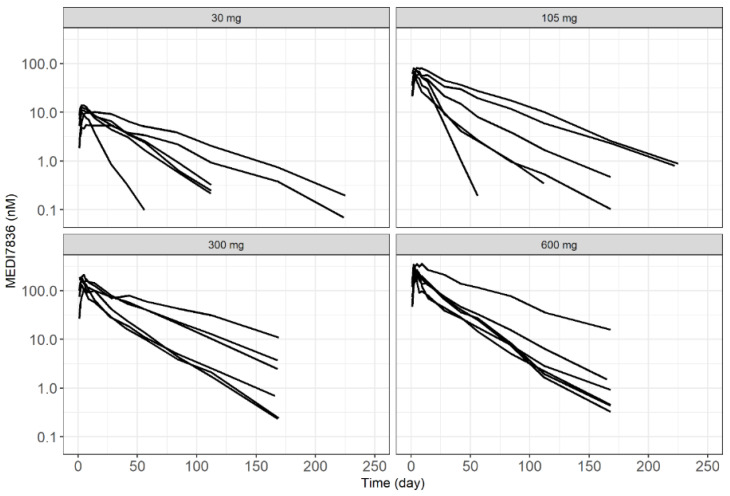
Observed individual serum concentration time-course of MEDI7836 from NCT02388347.

**Figure 3 pharmaceutics-13-00519-f003:**
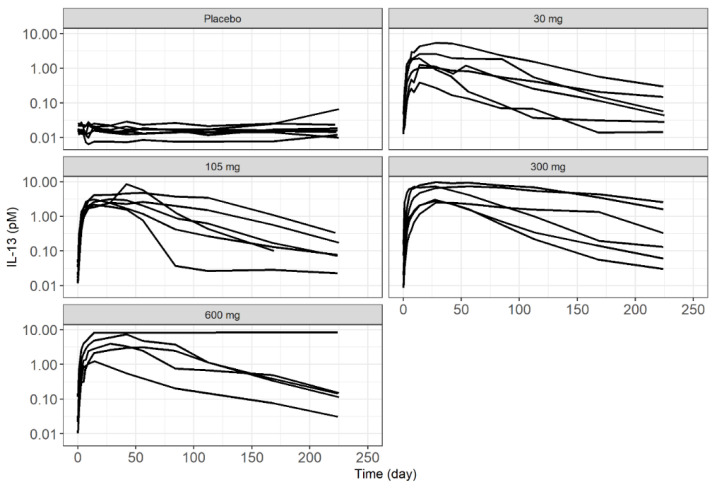
Observed individual serum measurements of IL13 time-course following subcutaneous administration of MEDI7836 or placebo from NCT02388347.

**Figure 4 pharmaceutics-13-00519-f004:**
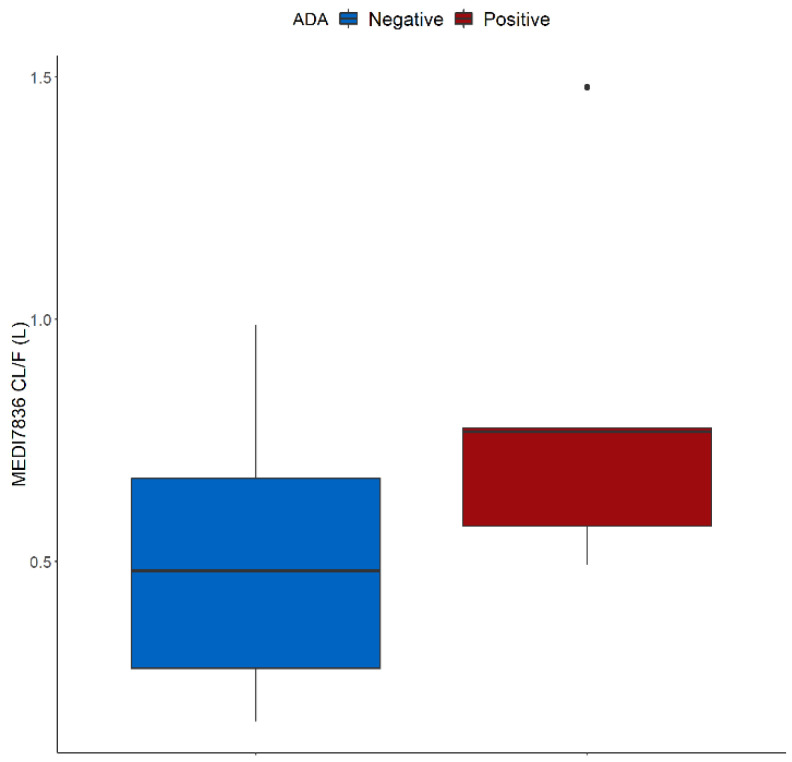
Impact of anti-drug antibodies on MEDI7836 apparent clearance rates (CL/F) in healthy volunteers (*n* = 24). ADA positive is defined as persistent positive post baseline (*n* = 16).

**Figure 5 pharmaceutics-13-00519-f005:**
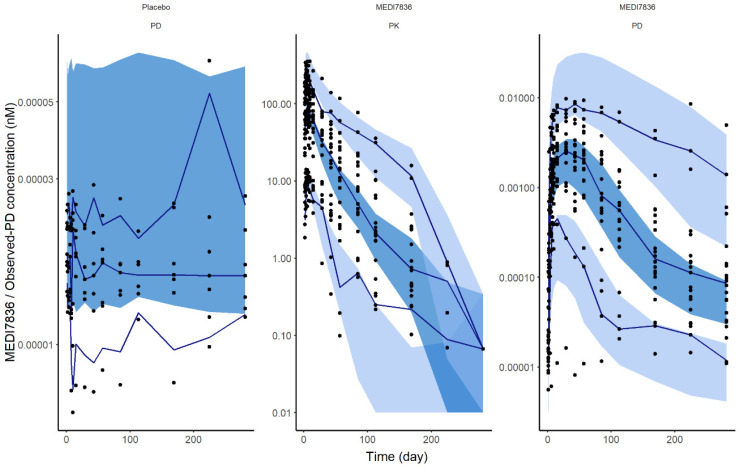
Standard visual predictive checks of the final PK-PD model of MEDI7836 based on 500 simulations (left panel: PD time-course in untreated control, central panel: MEDI7836 serum concentration time-course in treated groups (all doses), right panel: PD time-course in treated groups). Observed data are represented as dots, with median, 10th and 90th percentiles of the observed data represented by solid blue lines. The 95% confidence intervals based on corresponding percentiles of the simulated data are displayed as blue shaded areas, with the median prediction displayed in dark blue while extreme percentiles are displayed in light blue.

**Figure 6 pharmaceutics-13-00519-f006:**
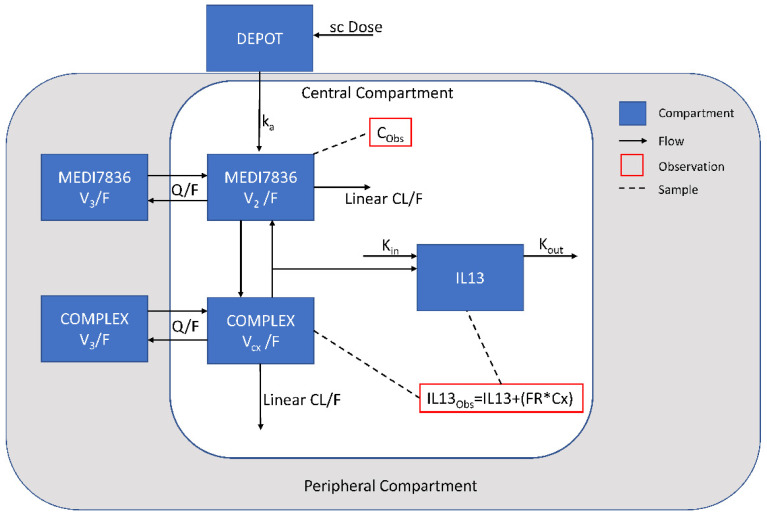
Schematic of the final population PK-PD model structure, Where SC, subcutaneous; CL/F, apparent linear clearance; k_a_, first order absorption rate; V_2_/F, apparent central compartment volume (MEDI7836); V_3_/F, apparent peripheral compartment volume (MEDI7836 and complex); V_cx_/F, apparent central compartment volume (complex); Q/F, apparent intercompartmental clearance rate of MEDI7836 and complex; K_in_, production rate of free IL13; K_out_, elimination rate of free IL13; FR, predicted fraction of the complex detected by the PD assay; C_Obs_, observed concentration of MEDI7836 in serum (PK); IL13_Obs_, estimated concentration of IL13 in serum assumed detected by the PD assay; IL13, free IL13 levels in serum; Cx, estimated concentration of MEDI7836:IL13 complex in serum.

**Figure 7 pharmaceutics-13-00519-f007:**
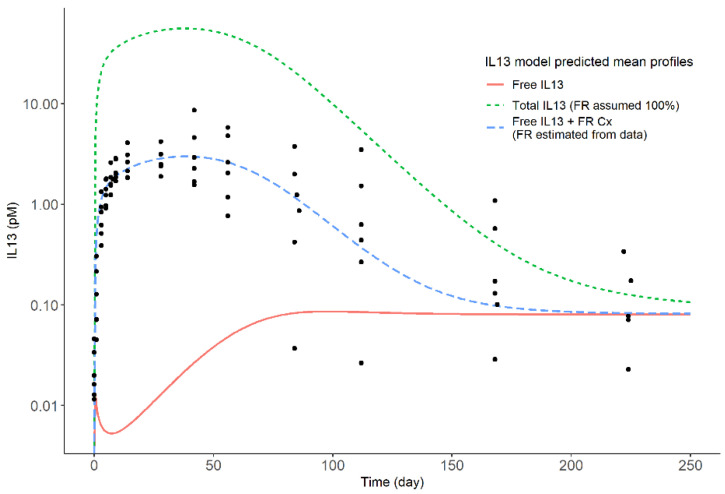
Comparison of individual time-course of observed serum IL13 (dots) and mean model predicted amount of free and total serum IL13 in the 105 mg dose group obtained from simulations based on the PK-PD model, which is illustrative of the observed PD in all treatment groups (see Figure 3). For comparison, the total IL 13 predicted by assuming all the MEDI7836:IL13 complex is captured by the bioanalytical PD assay is also shown.

**Figure 8 pharmaceutics-13-00519-f008:**
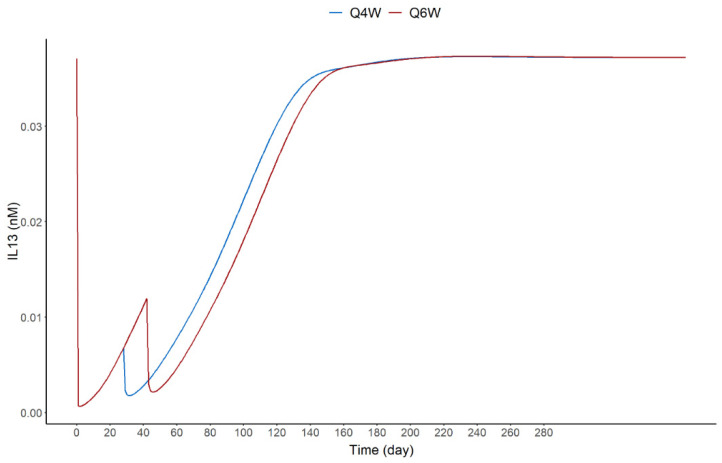
Dose simulations for MEDI7836 administered SC every 4 weeks (Q4W) and every 6 weeks (Q6W) for 2 dose events based on the final PK-PD model for a typical subject.

**Table 1 pharmaceutics-13-00519-t001:** Baseline demographics and anti-drug antibody status of subjects enrolled in NCT02388347 summarized by dose group, and pooled by active treatment and overall total.

Parameter	Placebo *n* = 8	MEDI7836 30 mg *n* = 6	MEDI7836 105 mg *n* = 6	MEDI7836 300 mg *n* = 6	MEDI7836 600 mg *n* = 6	MEDI7836 Total *n* = 24	Overall Total *n* = 32
Age (years)							
Median (IQR)	35.5 (25, 49)	38.5 (26, 45)	31.0 (21, 47)	29 (24, 47)	28.5 (18, 50)	35 (18, 50)	35 (18, 50)
Race (*n* (%N))							
Asian	0	0	0	1 (16.7%)	4 (66.7%)	5 (20.8%)	5 (15.6%)
Black or African American	1 (12.5%)	2 (33.3%)	2 (33.3%)	2 (33.3%)	1 (16.7%)	7 (29.2%)	8 (25.0%)
White	7 (87.5%)	4 (66.7%)	4 (66.7%)	3 (50.0%)	1 (16.7%)	12 (50.0%)	19 (59.4%)
Other	0	0	0	0	0	0	0
Body weight (kg)							
Median (IQR)	75.2 (62.4, 95.2)	81.2 (72.9, 84.2)	69.2 (60.4, 89.8)	81.0 (64.0, 91.6)	73.6 (67.4, 79.9)	76.5 (60.4, 91.6)	76.5 (60.4, 95.2)
ADA Response (*n* (%N))							
ADA-positive at baseline	2 (25.0%)	2 (33.3%)	4 (66.7%)	3 (50.0%)	3 (50.0%)	12 (50.0%)	12 (38%)
ADA-positive post-baseline	3 (37.5%)	5 (83.3%)	6 (100%)	5 (83.3%)	5 (83.3%)	21 (87.5%)	21 (66%)
ADA-positive post-baseline and positive at baseline	1 (12.5%)	2 (33.3%)	4 (66.7%)	3 (50.0%)	3 (50.0%)	12 (50.0%)	12 (38%)
ADA-positive post-baseline and not detected (or missing) at baseline	2 (25.0%)	3 (50.0%)	2 (33.3%)	2 (33.3%)	2 (33.3%)	9 (37.5%)	9 (28%)
Persistent Positive ^a^	1 (12.5%)	4 (66.7%)	4 (66.7%)	4 (66.7%)	4 (66.7%)	16 (66.7%)	16 (50%)
Transient Positive ^b^	2 (25.0%)	1 (16.7%)	2 (33.3%)	1 (16.7%)	1 (16.7%)	5 (20.8%)	5 (16%)

^a^; Persistent positive is defined as positive at ≥2 post-baseline assessments (with ≥16 weeks between first and last positive) or positive at last post-baseline assessment. ^b^; Transient positive is defined as negative at last post-baseline assessment and positive at only one post-baseline assessment, or at ≥2 post-baseline assessments (with <16 weeks between first and last positive). ADA = anti-drug antibody; IQR = interquartile range.

**Table 2 pharmaceutics-13-00519-t002:** Final population Pharmacokinetic/Pharmacodynamic model parameter estimates of MEDI7836. Parameter estimates are presented as the population median [5–95% Confidence Interval]. Confidence intervals were obtained from bootstrap analysis (*n* = 500) [19]. ^a^ Epsilon shrinkage for PK was 3.25%; ^b^ Epsilon shrinkage for PD was 6.65%.

PK Parameters	Estimate (95% Confidence Interval)	Shrinkage	PD Parameters	Estimate (95% Confidence Interval)	Shrinkage
CL/F (L/d)	0.441 [0.366–0.587]		K_in_ (nmol/d)	0.0173 [0.0136–0.0227]	
V_2_/F (L)	2.83 [2.21–4.06]		K_out_ (1/d)	180 [143–227]	
Q/F (L/d)	0.825 [0.638–1.13]		C_x_ fraction (%)	4.29 [2.98–5.96]	
V_3_/F (L)	8.03 [6.82–9.52]		V_cx_ (L)	13.6 [10.5–16.7]	
K_a_ (1/d)	0.156 [0.137–0.183]		ε3 (%CV) ^b^	6.9 [5.12–8.64]	
ADA CL Increase (%)	71.7 [17.3–155]		IIV of FR (%CV)	154 [64.1–295]	14.3%
ε1 (ug/L) ^a^	0.0546 [0.0131–13.9]		IIV of K_out_ (%CV)	57.1 [38.2–74.3]	17.6%
ε2 (%) ^a^	13.0 [11.1–36.1]				
IIV of CL/F (%CV)	53.3 [42.5–64.8]	12.3%			
IIV of V_2_/F (%CV)	72.7 [44.4–97.5]	17.8%			
IIV of Q/F (%CV)	69.6 [19.7–166]	28.6%			
IIV of V_3_/F (%CV)	46.2 [30.6–65.0]	23.6%			

## Data Availability

Data reported in this article may be obtained in accordance with AstraZeneca’s data sharing policy available at https://astrazenecagrouptrials.pharmacm.com/ST/Submission/Disclosure, accessed on 17 March 2021.

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
