# Peer review of "Pharmacokinetic-Pharmacodynamic Modelling of Systemic IL13 Blockade by Monoclonal Antibody Therapy: A Free Assay Disguised as Total"

_pharmaceutics, 2021, doi:10.3390/pharmaceutics13040519_

Round 1

Reviewer 1 Report

A theoretical PK-PD model designed by Hood et al. integrates independent knowledge about the assay characteristics to successfully elucidate apparently complex observations. Some minor issues should be addressed.

1. In my opinion, the word “pharmacokinetic” is suggested to include in the Title.

2. It is better to add an available link for Ref. 18.

3. Table 1: N = 7 in 300 mg group? N = 8 in 600 mg group? The statistical analysis of age and body weight of subjects in different groups could be performed.

4. The equation number is discontinuous in Page 9.

5. Figure 3: five serum measurements of IL13 time-course in 600 mg group?

6. Figure 5: Obviously, left panel: MEDI7836 serum concentration time-course in treated groups (all doses), centre panel: observed PD time-course in untreated control. Please revised the graph or legend.

Author Response

Thank you for your feedback on our paper. In response to your questions

  1. Pharmacokinetic added to paper title.
  2. Based on the reference style, links are not added to papers in this journal. For your information, the abstract is available from the PAGE website (https://www.page-meeting.org/default.asp?abstract=8976).
  3. The n=7 and n=8 are typos. Thanks for catching this. The display of demographic continuous variables is now displayed as median (IQR) in response to another reviewer
  4. Equation number corrected
  5. PD analysis for the 600 mg dose group included 5 patients
  6. Plot changed.

Reviewer 2 Report

It's a very well written paper by an international group of PK/PD industrial experts. The PK/PD modeling approach is quite interesting and deserves to be published.

I'm a bit surprised that NCT02388347 trail was placebo-controlled; being a classical, dose-finding phase I trial on healthy volunteers, it should be a single-arm study. What about MEDI7836 DLT and MTD? Why have you not enrolled patients triplets?

When dealing with small and asymmetric sample size (like the yours), continuous covariates are to be reported only as median(IQR) and not mean/sd

line 53: dupilumab targets both IL4 and IL13, plese correct this typo.

Author Response

Thanks for your feedback on our paper. In response to your questions

1. A control arm was included because we were investigating the pharmacodynamic effects of MEDI7836 on IL13, and in order to understand the IL13 profile and distinguish drug effect from normal IL13 behaviour a placebo arm was necessary.

2. The continuous variables in table 1 have been reported as median (IQR) in response

3. The dupilumab information on line 53 has been corrected.

Reviewer 3 Report

The authors presented a paper on a Population PK-PD model approach for data obtained in clinical Phase I studies after single-ascending SC doses of the IL13 neutralizing mAB MEDI7836 in healthy adult male subjects.

This model was developed in order to understand the unexpected behaviour of free IL13 observed after administration of MEDI7836, and proposes that the observed behaviour suggests that, although the analytical method was validated for quantifying free IL13, total IL13 was being determined. The major relevance of the paper is trying to explain an observed experimental data that do not follows the “expected” behaviour. However, its major flaw is the fact that no experimental confirmation was made at the analytical method pos-validation in order to confirm the model based explanation and solve the issue for future use. The data also is short in number (only 24 PK subjects), and complexity (only single dose administration, no IV administration) that limit the applicability of the model itself.

Regarding the method performance evaluation, the GOF plots are lacking and should be presented. The VPCs are lacking the 2.5% and 97.5% lines and areas. Assuming that the presented area is related to the simulated median, the initial portion of the PK data do not seem to be well capture.

Shrinkage was not presented for the model parameters. In addition, the investigation of covariate relationships is not well defined and it cannot be currently assessed if properly performed.

ADA positive post-baseline status was statistically associated with faster MEDI7836 mAb clearance. However, 21 of 24 subjects (87.5%) were positive for this covariate. The relevance of this finding should be discussed. Since only 3 subjects were ADA negative, the authors should consider changing Figure 4 to, at least, present individual data points.

The Vc and Vp values, as the authors also agreed, are not entirely according to what expected. Beside the already presented explanations, the authors should also discussed the quality of the data as very high IIV in Vc/F (72.7%) and Q/F (469.6%!) was described.

In page 5, line 148, authors referred that 13.9 PK samples and 14.9 PD samples composed the dataset. Is this correct?

Author Response

  1. The limitations of this work are addressed in lines 395 – 402. NCT02388347 was designed by a multidisciplinary team to primarily assess safety and tolerability of MEDI7836. The PD efficacy of MEDI7836 was an exploratory objective. Based on the poor PK of MEDI7836, the project was closed. MEDI7836 clinical development has been terminated by the organisation, therefore there was no business justification for investigating the observed PD.
  2. GOF plots have been added to the supplementary information, specifically; IPRED vs DV for PK and PD, CWRES vs IPRED for PK and PD, and CWRES vs Time for PK and PD.
  3. ETA Shrinkage was added to IIV parameters, and Eps shrinkage was added to PK and PD BSV parameters on table 2.
  4. Persistent ADA positive status was shown affect CL. 16 patients out of 32 were shown to have persistent ADA status. The ADA status used has been clarified in the text. N numbers have been added the legend on figure 4, clarifying the amount of ADA positive patients.
  5. The high variability on Q/F was a typo. The correct value was 69.6%. The high observed IIV has been discussed in lines 325-326 of the document. High IIV is common in apparent parameters as they are representative of a ratio of parameters, specifically, bioavailability and CL, Q, and V. A limitation of this study design was the lack of the IV dose which would have allowed us to more accurately estimate distributional parameters.
  6. The 13.9 and 14.9 were the average number of samples per patient for PK and PD, respectively. The text has been updated to remove this ambiguity.

Reviewer 4 Report

The manuscript describes a novel approach in modeling mAb PK and PD (total target) data. Having experience in modeling mAb PK/PD data, this manuscript was at peak of my interest and I believe it merits publication. I strongly recommend the authors to share the NONMEM script as a supplementary material, which would allow reproduction and adaptation of the work. This is because only the model schematic and parameters are available in the manuscript. In some occasions, there may be misunderstanding of the schematic leading to difficulties in reproducing the model. If the NONMEM script contains any confidential information, those specific information can be removed from the script before publication.

Author Response

Thanks for your feedback on our paper. We will include the NONMEM script as supplementary data

Round 2

Reviewer 3 Report

The authors have resolved all my previous comments.